# Effects of Scanning Speed on the Polished Surface Quality of Mold Steel by Dual-Beam Coupling Nanosecond Laser

**DOI:** 10.3390/ma16041477

**Published:** 2023-02-09

**Authors:** Huihui Zhang, Xiaoxiao Chen, Wenwu Zhang, Dianbo Ruan

**Affiliations:** 1School of Mechanical Engineering and Mechanics, Ningbo University, Ningbo 315211, China; 2Ningbo Institute of Materials Technology and Engineering, Chinese Academy of Sciences, Zhejiang Provincial Key Laboratory of Aero Engine Extreme Manufacturing Technology, Ningbo 315201, China; 3University of Chinese Academy of Sciences, Beijing 100049, China

**Keywords:** coupled nanosecond laser polishing, S136D mold steel, laser cutting, surface roughness, polished surface quality

## Abstract

In this paper, a novel dual-beam coupled nanosecond laser was used to polish S136D mold steel. The effects of scanning speed, total fluence, spot overlap ratio, and *SPS_N_* on surface quality were analyzed. The polished surface roughness Ra without ultrasonic cleaning is too large due to slag, splash, and dust produced by laser polishing. When scanning speed is 1250 mm/min, surface roughness Ra with ultrasonic cleaning is reduced from the original surface 1.92 μm to 0.72 μm, and the surface roughness Ra is reduced by 62.50%. When the *F_tot_* is 35.38 J/cm^2^, the minimum value of surface roughness Ra is 0.72 μm. If the total fluence is higher or lower, it is not conducive to reducing the surface roughness, the total fluence is higher, and there is a polished surface with SOM phenomenon. The polished surface with spot overlap ratio of 98.55% has a smooth morphology, and a minimum value of surface roughness Ra of 0.41 μm. When the specimen is inclined at a certain angle, the high-magnification camera captures color on the polished surface. It is found that the microscopic texture of molten material flow trace and polishing scanning track is obvious. Polished surface is mainly distributed with Fe, Cr, C, and O elements. The surface material processing speed per unit time is low, and the polishing surface quality is improved less. The maximum surface roughness Ra is 1.98 μm. The minimum Ra of polished surface with smoother morphology is 0.41 μm, and surface profile height is basically the same. The research results show that the new dual-beam coupled nanosecond laser polishing technology can improve surface quality of materials. This research work provides process guidance for laser polishing effect analysis and mechanism innovation.

## 1. Introduction

Mold is an indispensable basic process equipment in the manufacturing industry. The surface quality of the mold has a great influence on the performance, service life, and cycle of products [1]. The polishing of mold steel is mainly focused on high efficiency and high precision direction, and many molds require surface roughness Ra to reach the nanometer level [2]. S136D mold steel has good corrosion resistance, high hardness, and polishing performance, which is widely used in plastic products of molds. The molds manufactured by traditional polishing technology gradually cannot meet the production requirements of high precision and high-performance molds [3]. Traditional mold polishing technology has the following limitations: limited speed, high cost, poor working environment, time-consuming, and labor-intensive [4]. As a new polishing technology, compared with the traditional polishing technology, laser polishing has the advantages of green and environmental protection, precision and ultra-precision polishing, non-contact polishing, and is a simple process [5,6,7]. Bordatchev et al. [8] pointed out that laser polishing can reduce the manual polishing processing time by an order of magnitude. Since the 1980s, laser polishing technology has attracted the research of scholars at home and abroad. In 1983, Xiao et al. [9] used a 300W CO_2_ laser to polish BK-7 and Zerodur glass, and cracks appeared on the surface of the glass after polishing. In 1986, Tuckerman et al. [10] proposed the use of pulsed laser polishing of gold and aluminum materials. Since the 2000s, Ramos et al. [11] conducted the first comprehensive study on laser polishing of metallic materials, which showed laser polishing divided into thermal polishing and cold polishing [12], and laser polishing metallic materials mostly belonging to thermal polishing. Ramos and Bourel [13] proposed that thermal polishing could be divided into Surface Shallow Melting (SSM), and Surface Over Melting (SOM), according to the different mechanisms of action in the laser polishing process. 

In recent years, a large number of scholars have researched the effects of fluence [14], scanning speed [15], defocus amount [16], laser incidence angle [17], and pulse laser radiation [18] on the laser polishing effect. Jaritngam et al. [19] studied nanosecond laser polishing of Ti6Al4V titanium alloy, which showed that when laser fluence is 1~3 J/cm^2^, surface roughness Sa decreased by 43%. Chow et al. [20] explored the effect of different defocus amounts on the surface roughness of polished mold steel, and the results showed that when the defocus amounts was greater than 2.2 mm, the surface roughness of AISI H13 mold steel reduced by 39.7%. Pfefferkorn et al. [21] pointed out that the beam diameter of laser affects the surface roughness of polishing S7 tool steel. Chen et al. [22] explored polishing ASP23 mold steel with different scanning paths, and found that using complex scanning paths can significantly reduce the surface roughness of ASP23 mold steel. Laser polishing can be divided into CW laser polishing and pulsed laser polishing according to the laser output mode. CW laser is usually used for rough polishing, while pulsed laser is mainly used for fine polishing [23,24]. Zhou et al. [25] used CW laser to polish S136D mold steel, and designed L16 (44) orthogonal experiment to explore the change of surface roughness of S136D mold steel with fluence, and the results showed that the surface roughness Ra decreased from 5.358 µm to 0.764 µm. Chang et al. [26] studied the CW laser polishing of SKD61 tool steel with a wavelength of 1070 nm, and found that polishing SKD61 tool steel with higher fluence produced surface ripples. Folwaczny et al. [27] polished In-Ceram Spinell with 308 nm XeCl excimer pulsed laser by changing fluence. The results showed that polished surface roughness Ra decreased from 4.14 μm to 1.13 μm, when the fluence is 6.28 J/cm^2^. Mikheev et al. [18] studied the possibility of the pulse laser radiation treatment of thin metal films on glass substrates, laser irradiation may affect the improvement of laser polishing effect. Traditional laser polishing mainly uses a single laser beam for polishing, and in recent years, some scholars have studied the dual laser polishing system. Nüsser et al. [28] developed a dual laser polishing system, which was composed of CW laser and pulsed laser, and found that the surface of Ti6Al4V titanium alloy polished by dual laser is smoother than that polished by single laser. Zhou et al. [29] polished S136H tool steel with double laser, and the results showed that surface roughness Sa of S136H tool steel decreased from 0.877 μm to 0.142 μm. Ukar et al. [30] studied the polishing of DIN 1.2379 tool steel by CO_2_ and high-power diode laser (HPDL) CW laser, and surface roughness decreased by 90% after polishing. Liu et al. [31] explored the mixed polishing by CW and pulsed laser, and surface roughness of Inconel 718 superalloy decreased from 15.75 μm to 0.23 μm. 

In conclusion, most scholars have carried out a series of researches on the selection of process parameters that affect the laser polishing effect. A few scholars have studied the polishing by dual laser polishing system, which was composed of pulsed laser and CW laser. At present, most of the researches focus on the polishing of materials by lasers without beam modulation. The selection of process parameters, surface morphology, polishing mechanism, and element content change of coupled laser polishing of metal materials needing further study are rarely reported. In this paper, a new type of dual-beam coupled nanosecond laser is used to polish mold steel. The effects of total fluence, spot overlap ratio, and surface process speed on the surface quality of mold steel were analyzed by combining with a polishing mechanism. Laser Scanning Confocal Microscope (LSCM, model: KEYENCE VK-X200K), Scanning Electron Microscope (SEM, model: FEI Quanta FEG 250), and High Magnification Camera (HMC, model: AM7915MZY) were used to observe the polished surface. Energy Dispersive Spectroscopy (EDS, model: FEI, Hillsboro, OR, USA) was used to analyze the changes of surface element content. This work mainly focuses on the improvement of polished surface quality. The effect of scanning speed on polished surface quality was analyzed from a different point of view, which could provide technological guidance for laser polishing effect analysis and mechanism innovation. 

## 2. Materials and Methods

### 2.1. Experimental Materials

The experimental material is S136D mold steel with a size of 20 mm × 20 mm × 5 mm, and the chemical composition is shown in Table 1.

In order to remove surface dust and stains, the mold steel was placed in a beaker containing alcohol solution for ultrasonic cleaning for 8 min before the polishing experiment.

### 2.2. Experimental Methods

The laser in the D-200CNC laser processing system adopts a nanosecond pulsed green laser, with an adjustable output frequency of 0.5~100 kHz, pulse duration of 78 ns, wavelength of 532 nm, maximum output power of 23 W, and the beam quality (M^2^) less than or equal to 1.5. The beam emitted by the laser reaches the spectroscope through the 45° reflector mirror, the spectroscope divides the single beam into two laser beams. The beam 1 and beam 2 are coupled into a spot through the focusing mirror, and the movement of the CNC machine adjusts the coupled spot to act on the surface of the S136D mold steel and dual-beam coupled nanosecond laser processing system device, as shown in Figure 1.

When the laser power is 1.0 W, the repetition frequency is 10 kHz, the number of scanning is 1, and the scanning pitch is 40 μm, the coupling beam waist radius is calculated as 43.11 μm. According to the polished area planning, the polished experiment is completed on the mold steel by changing the scanning speed *v*_s_ (mm/min), as shown in Figure 2a. The beam 1 and beam 2 are coupled into a beam through the focusing mirror, the coupling beam obeys the energy distribution of Gaussian beam, and there is no change in the properties of the coupled beam. “The included angle” is the value of the angle between beam 1 and beam 2, which is 37.30°, the orientation of the plane of incidence s relative to the velocity vector *v*_s_ is shown in Figure 2b. “The included angle” modulated by the optical path is ideal and can be directly used to study the processing quality of materials. The experimental parameters of variable speed polishing mold steel are shown in Table 2. The polished surface was observed by using LSCM, SEM, and HMC. 

The middle position of the polished surface is selected for three measurements to obtain the average roughness Ra, as shown in Figure 3. 

## 3. Results and Discussion

### 3.1. Effects of Scanning Speed on Polishing Quality

(1)Analysis of surface roughness and morphology

Figure 4 shows the relationship between the surface roughness Ra of mold steel and scanning speed. As shown in Figure 4, the measurement roughness Ra is too large because of slag, splash, and dust on the polished surface before cleaning. As shown in Figure 4a, surface roughness Ra first increases and then decreases with the increase of scanning speed. When scanning speed is 250 mm/min, the minimum value of surface roughness Ra is 1.31 μm, and surface roughness Ra decreases by 31.77%. From Figure 4b, it can be concluded that the surface roughness first decreases with the increase of scanning speed. When the scanning speed is 1250 mm/min, the minimum value of surface roughness Ra is 0.72 μm, surface roughness Ra reduces by 62.50%, and then surface roughness Ra increases.

During laser irradiation on the material surface, when the laser heat energy on the material surface reaches a certain level, the crests of material surface reach the melting point first and then melt. The increase of scanning speed will accelerate the flow of molten material, the molten material flows to the troughs under the action of surface tension and gravity, the troughs are filled, and the surface roughness of material is smoothed [29]. This is the diagram of the surface roughness reduction process, as shown in Figure 5. Compared with single beam, the spot diameter of the coupling beam is adjustable, the energy distribution at the beam waist is variable, and the spot of coupling beam is elliptical. The theoretical radius of single beam is higher than the coupling beam waist radius which is calculated as 43.11 μm. The laser fluence of coupling beam is larger than that of single beam under the same laser power and pulse frequency. The spot of coupling beam is elliptical, which has the advantage of a large processing area. Compared with single beam laser processing, the coupled beam laser processing has more adjustable process parameters and a wider process range. Different spot diameters of the coupling beam are obtained by adjusting the angle between two beams, which meets the needs of processing capability. With the increase of the angle between the two beams, the eccentricity of the elliptical spot increases. 

During variable scanning speed polishing mold steel, changing the scanning speed will affect the spatial energy input. The spatial energy can be described by the total fluence (*F_tot_*), and the formula of the total fluence is expressed as Equation (1) [32]:(1)$$F_{tot}=\frac{F_{0}w_{0}f}{v_{s}}\left[\frac{J}{cm^{2}}\right]$$
where *F*_0_ is the laser fluence (J/cm^2^), *w*_0_ is the beam waist radius (μm), *f* is the laser repetition frequency (Hz), *v_s_* is the scanning speed (mm/min).

The relationship between the total fluence (*F_tot_*) and scanning speed is presented in Figure 6, according to experimental parameters and Equation (1). It can be concluded from Figure 6 that the total fluence decreases with the increase of scanning speed. The relationship between surface roughness Ra before and after cleaning and total fluence, which can be obtained by combining Figure 4 and Figure 6. When the *F_tot_* is 88.46 J/cm^2^, the surface roughness Ra increases to a maximum value of surface roughness of 1.73 μm. When the *F_tot_* is 176.92 J/cm^2^, the minimum value of surface roughness Ra is 1.31 μm. It can be seen from Figure 4b, that when the *F_tot_* is 35.38 J/cm^2^, the minimum value of surface roughness Ra is 0.72 μm. It can also be seen if the total fluence is higher or lower, and that it is not beneficial to reducing surface roughness. When the total fluence is lower, the surface material cannot be fully melted, and the original gully on the surface of the material is not removed. When the total fluence is constant, the crests of surface material are melted during the laser polishing process. When the molten material flows to the trough under the action of surface tension and gravity, the troughs of material surface are filled, the surface roughness is reduced, and SSM phenomenon is formed, which is the evolution process of the SSM mechanism [33], as depicted in Figure 7a. When the laser acts on the material surface, the crest part of the material surface first reaches the melting point and begins to melt. The trough part of the material surface is in the unmelted state. The molten material flows to the trough due to gravity and surface tension, and finally obtains a smooth surface. t_a_ represents the laser which acts on the trough part of the material, t_b_ and t_c_ represent the molten material flows to the trough part, and t_d_ indicates the final smooth surface. The schematic diagram of SSM mechanism evolution is drawn according to the time sequence t_a_→t_b_→t_c_→t_d_, as shown in Figure 7. When the total fluence is higher, excessive surface material melting during the laser polishing process will remove the original surface bulges. However, the surface tension generated by different material states inside and outside the laser spot will cause surface ripples and gullies, which will lead to the increase of surface roughness, resulting in SOM phenomenon. The evolution process of SOM mechanism [34] is depicted in Figure 7b. 

The spot overlap ratio (*SO_R_*) and trajectory overlap ratio (*TO_R_*) during laser polishing, as shown in Figure 8, which both determine the ablation range and the superposition of thermal stress. In order to achieve a better polishing effect, choosing a reasonable scanning speed and scanning pitch is necessary. The formula for the spot overlap ratio and trajectory overlap ratio is expressed as Equations (2) and (3) [19]:(2)$$SO_{R}=1-\frac{v_{s}}{2w_{0}f}$$
(3)$$TO_{R}=1-\frac{\Delta y}{2w_{0}}$$
where ∆y is the scanning pitch (mm), *w*_0_ is the beam waist radius (μm), *v_s_* is the scanning speed (mm/min), *f* is the laser repetition frequency (Hz).

Before the variable speed polishing mold steel experiment, the waist radius of the coupled beam was calculated as 43.11 μm, according to the experiment of dual-beam coupled nanosecond laser ablation mold steel with variable power. The spot overlap ratio (*SO_R_*) can be obtained from the experimental parameters and Equation (2), as presented in Table 3.

Figure 9 and Figure 10 show the three-dimensional morphologies of the polished surface at spot overlap ratios of 99.52%, 98.55%, 97.58%, and 96.62%. Figure 9 shows that the surface morphologies are uneven, and there are apparent crests and troughs, and the polished surface with spot overlap ratio of 97.58% has obvious depressions. It can be seen from Figure 10 that there are few crests and troughs on the polished surface, the overall surface morphologies tend to be flat, and the surface flatness is improved. The polished surface morphology with spot overlap ratio of 98.55% is the smoothest. It can be seen that surface morphology after cleaning has been improved to a certain extent, and the roughness has decreased by combining as shown in Figure 9 and Figure 10. The height difference between crest and trough of the polished surface becomes smaller, and the surface morphology is flat because of the flow of molten material in the polishing process.

Figure 11 shows the polished surface observed through HMC. Through Figure 11a, it can be seen that the polishing marks and polishing area are obvious; S136D mold steel mainly contains Fe, Cr, C, and Si elements. The laser polishing process is accompanied by a strong oxidation reaction, which may produce light green to dark green Cr_2_O_3_, white SiO_2_, reddish brown Fe_2_O_3_, black Fe_3_O_4_, and other oxides; these oxides with obvious colors are on the polished surface, and the colorful polished surface can be captured by using HMC, when the specimen is inclined at a certain angle.

(2)Analysis of surface micro-morphology and element content

The polished surface morphology was observed by SEM, and the element content and element distribution map were detected by EDS, as seen in Figure 12 and Figure 13. Figure 12b–d show the surface morphologies of different polishing areas obtained at the same magnification, which has obvious surface bulges, flow traces of molten material, and polishing scanning trajectory. The surface element of polished Zone 1, Zone 2, and Zone 3 are mainly Fe, Cr, and C elements, according to Figure 12b-1–d-1. The average content of Fe, Cr, and C elements is 80.46%, 14.78%, and 2.78%, respectively. The content of Fe element on the polished surface decreases, and the content of C element increases, compared with the content of Fe and C element on the original surface. The increase of the content of C element improves the hardness of steel, but the plasticity, toughness, and corrosion resistance of steel will become worse. When scanning speed is 250 mm/min, the total fluence of polishing is the largest, and the surface of the material heats up sharply to produce molten material, which smooths the irregular surface. The polishing process is accompanied by a strong oxidation reaction. It can be seen from Equation (4) the increase of the content of C element on the polished surface. In Figure 13a–c, the main concentration regions of Fe and O elements are marked. From Figure 13a-1–c-1, it can be seen that the polished surface is mainly distributed with Fe, Cr, C, and O elements. The polished surface mainly contains Fe, Cr, C, O, and Si elements. The content of O element is high in the flow trace area of the molten material. The strong oxidation reaction during the polishing process will cause the content of O element to increase.
(4)$$2\mathrm{Fe}+3{\mathrm{CO}}_{2}={\mathrm{Fe}}_{2}O_{3}+3\mathrm{CO}$$

It can be seen from Figure 14 that there were obvious polishing marks on the polished surface, and the scanning path of the coupled beam polishing adopts the bow shape. When scanning speed is 250 mm/min, the depth of the molten pool on the polished surface is deeper, and the surface is more likely to generate polishing marks. The molten material flow trace can be clearly observed at high magnification. 

### 3.2. Analysis of Relationship between Surface Process Speed and Polished Quality

The polished surface quality is evaluated by surface roughness, three-dimensional surface morphologies, surface micro-morphologies, and element content change. Polished quality can be quantified by surface roughness. The lower the surface roughness, the better the polished surface quality. The surface process speed (*SPS_N_*) is one of the evaluation indexes of polishing effect, the *SPS_N_* is expressed as the area of material removed by a single scan per unit time. The formula is presented as Equation (5) [32]:(5)$$SPS_{N}=\frac{dA}{dt}=\frac{v_{s}∆y}{N}\left[\frac{cm^{2}}{min}\right]$$
where ∆y is the scanning pitch (mm), *v_s_* is the scanning speed (mm/min), *N* is the scanning times.

The calculation formula for improvement percentage of material surface roughness *Ra* is presented as Equation (6) [35]:(6)$$\mathrm{Reduction}\left(\%\right)=\frac{Ra_{original}-Ra_{\mathrm{polished}}}{Ra_{original}}\times 100\%$$
where Ra_original_ is the original surface roughness Ra (μm), Ra_polished_ is the polished surface roughness Ra (μm).

The *SPS_N_* can be obtained through Equation (5), as presented in Table 4.

Figure 15 shows the relationship between the surface roughness Ra reduction percentage and *SPS_N_* is obtained according to Equation (6). It can be seen from Figure 15 that surface roughness Ra reduction percentage firstly increases with the increasing surface process speed, and the maximum reduction of surface roughness Ra is 62.50%, when the *SPS_N_* is 5 cm^2^/min. When the *SPS_N_* is low, the improvement of the polished surface quality is less. When the *SPS_N_* increases to a certain value, the troughs are filled, and the surface roughness of material is reduced, SSM phenomenon occurs on the polished surface. When the surface process speed increases, the surface material cannot be fully melted to smooth out the irregular profile, and SOM phenomenon occurs on the material surface, which leads to the reduction of polished surface quality. Therefore, if the *SPS_N_* is higher or lower, it is not conducive to the improvement of polished surface quality. 

The original surface and polished surface of mold steel were observed by LSCM, and the surface roughness Ra; three-dimensional morphology and surface profile were obtained, as shown from Figure 16, Figure 17 and Figure 18. It can be seen from Figure 16a that there are obvious crests and troughs on the original surface, and the surface profile is very irregular. The maximum value of the original surface roughness Ra is 1.98 μm. Figure 16b shows that the original surface profile fluctuates greatly, and the surface crests and troughs are obvious. The height difference between crest and trough is large, and the original surface roughness is large.

Figure 17 shows that when the *SPS_N_* is 1 cm^2^/min, the surface roughness Ra decreases from 1.98 μm to 1.68 μm, and surface roughness Ra reduces by 15.15%. After laser polishing, the surface height tends to be consistent, and the height difference between crests and troughs decreases, but the bumps and gullies on the polished surface increase, resulting in worse three-dimensional morphology of the polished surface and affecting the improvement of polished surface quality. 

Figure 18a shows that when the *SPS_N_* is 3 cm^2^/min, the minimum value of surface roughness Ra is 0.41 μm, and when the roughness Ra decreases from 1.98 μm to 0.41 μm, surface roughness Ra reduces by 78.66%, and the surface is smooth without obvious bumps and gullies. Figure 18b shows that the profile fluctuation of the polished surface is basically the same, and the height difference between crests and troughs is greatly reduced. The polished surface is smoother than the original surface and better surface morphology is produced, which shows that coupled nanosecond laser polishing can significantly improve the surface quality of metal materials.

## 4. Conclusions

In this paper, the experiments of dual-beam coupled nanosecond laser polishing of mold steel were carried out. The effects of scanning speed, total fluence, spot overlap ratio, and surface process speed on the polished surface roughness, morphology, and elemental content before and after cleaning were analyzed, and the mechanism of laser polishing was also discussed. The main conclusions are as follows.

(1) In variable scanning speed polishing mold steel, the measurement roughness Ra before cleaning is too large because of slag, splash, and dust on the polished surface before cleaning. When scanning speed is 1250 mm/min, the surface roughness Ra after cleaning decreases from 1.92 μm to 0.72 μm, with a decrease of 62.50%. When the *F_tot_* is 35.38 J/cm^2^, the minimum value of surface roughness Ra is 0.72 μm. If the total fluence is higher or lower, it is not conducive to improvement of surface quality. When the total fluence is higher, SOM phenomenon exists on the polished surface.

(2) Comparing the polished surface morphology before and after cleaning, the flatness of the cleaned surface was improved. When the spot overlap ratio is 98.55%, the polished surface is smoothest, and the minimum value of surface roughness Ra is 0.41 μm. When the specimen is inclined at a certain angle, the polished surface showed obvious color changes as captured by the high magnification camera. Through SEM observation, it is found that the microscopic texture of molten material flow trace and polishing scanning track is obvious, when there is a scanning speed of 250 mm/min. The EDS detection finds that the polished surface is mainly distributed with Fe, Cr, C, and O elements.

(3) The surface material process speed is lower, the improvement of the polished surface quality is worse. There are more crests and troughs on the original surface, and the surface roughness Ra is larger. When the *SPS_N_* is 1 cm^2^/min, the maximum value of polished surface roughness Ra is 1.68 μm, and the surface bumps and gullies increase. When the *SPS_N_* is 3 cm^2^/min, the minimum value of polished surface roughness Ra is 0.41 μm, and the surface morphology is smoother.

(4) The research of polishing technology by using the novel dual-beam coupled nanosecond laser can provide mechanism and process guidance for the analysis of laser polishing effects.

## Figures and Tables

**Figure 1 materials-16-01477-f001:**
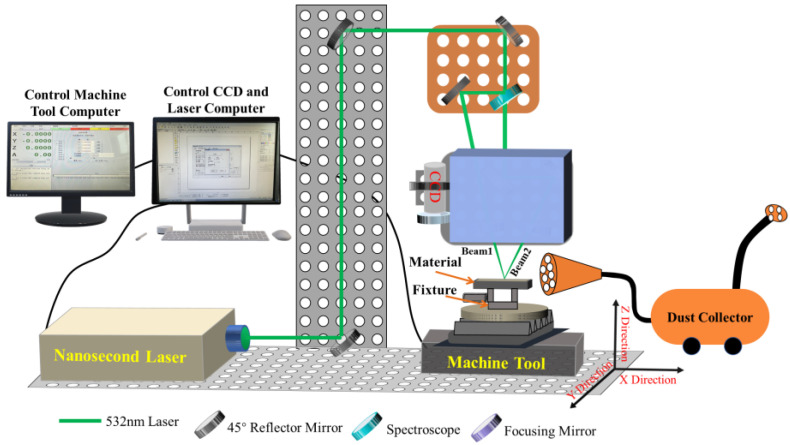
Schematic diagram of dual-beam coupled nanosecond laser processing system.

**Figure 2 materials-16-01477-f002:**
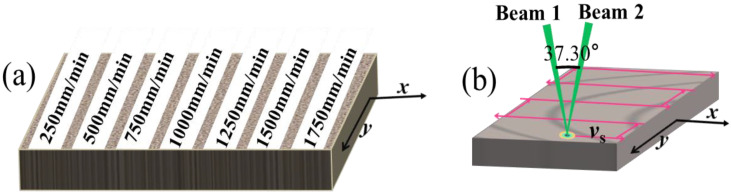
Schematic view of scanning trajectory and polished area planning. (**a**) The polished area planning; (**b**) The scanning trajectory.

**Figure 3 materials-16-01477-f003:**
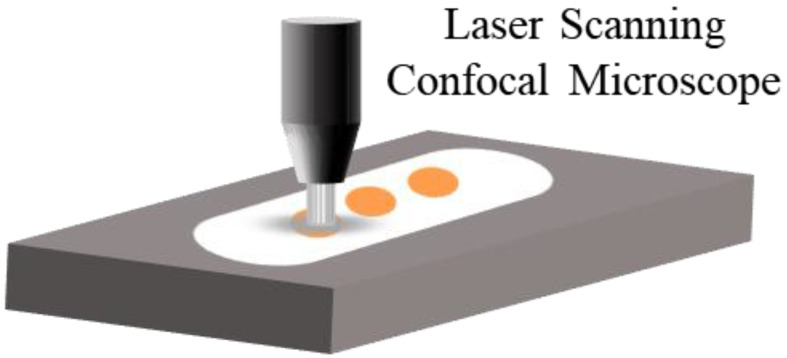
Measurement method of polished surface by LSCM.

**Figure 4 materials-16-01477-f004:**
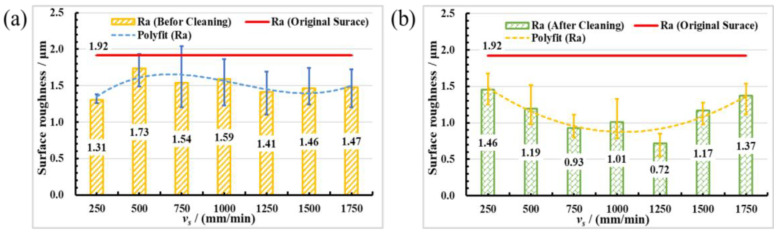
Variations of surface roughness of before and after cleaning with the laser scanning speed. (**a**) Before cleaning; (**b**) After cleaning (measured by LSCM at 1000× magnifications).

**Figure 5 materials-16-01477-f005:**
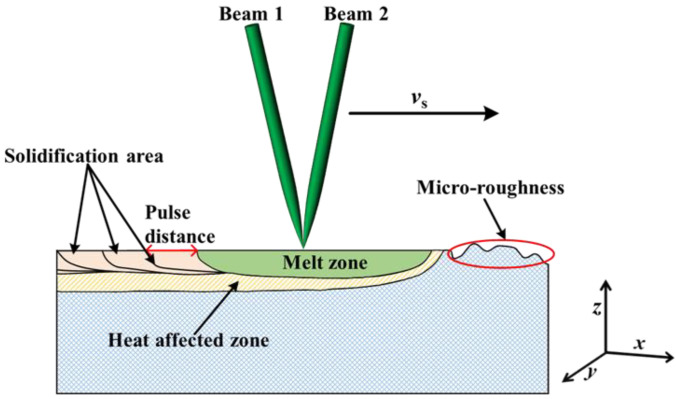
Schematic process principle of dual-beam coupled nanosecond laser polishing.

**Figure 6 materials-16-01477-f006:**
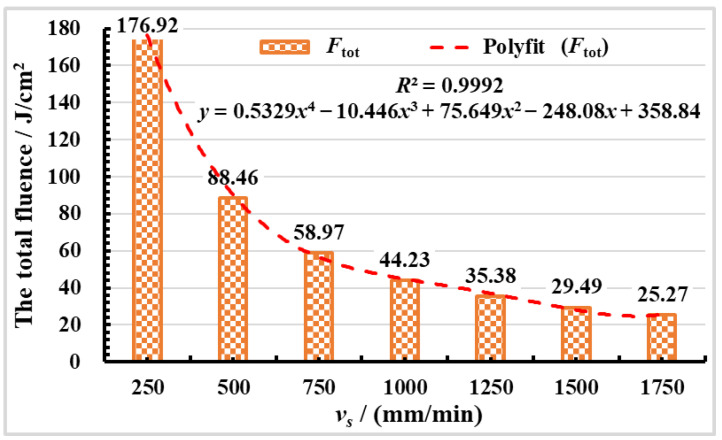
Variations of the total fluence with laser scanning speed.

**Figure 7 materials-16-01477-f007:**
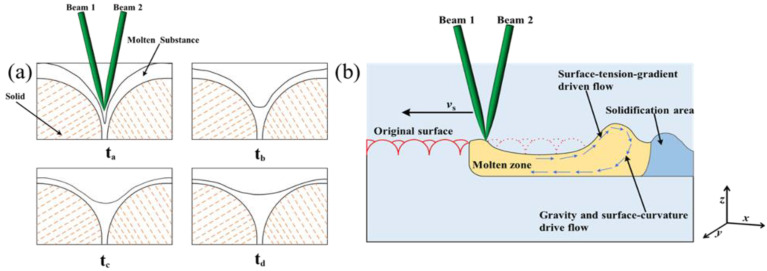
Schematic diagram of mechanism evolution of the SSM and SOM. (**a**) SSM, with the time sequence of t_a_, t_b_, t_c_, t_d_; (**b**) SOM.

**Figure 8 materials-16-01477-f008:**
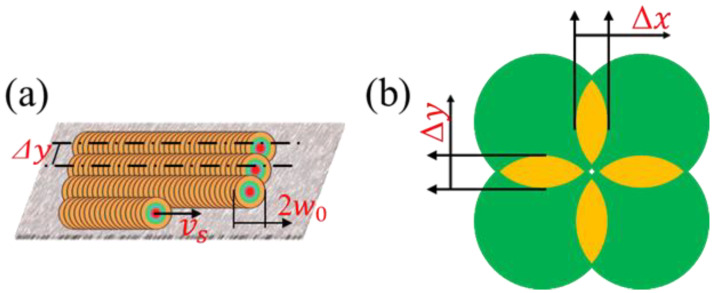
Diagrams of the dual-beam coupled nanosecond laser polishing of mold steel. (**a**) Trajectory overlap; (**b**) Spot overlap.

**Figure 9 materials-16-01477-f009:**
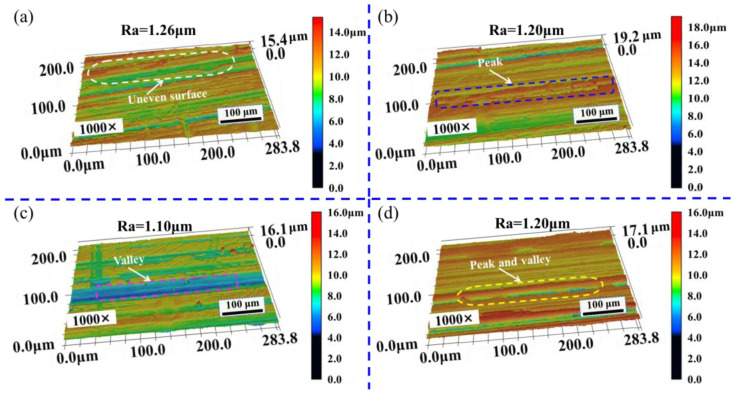
Three-dimensional morphologies of polished surface under different spot overlap ratios before cleaning. (**a**) 99.52%; (**b**) 98.55%; (**c**) 97.58%; (**d**) 96.62% (observed by LSCM at 1000× magnifications).

**Figure 10 materials-16-01477-f010:**
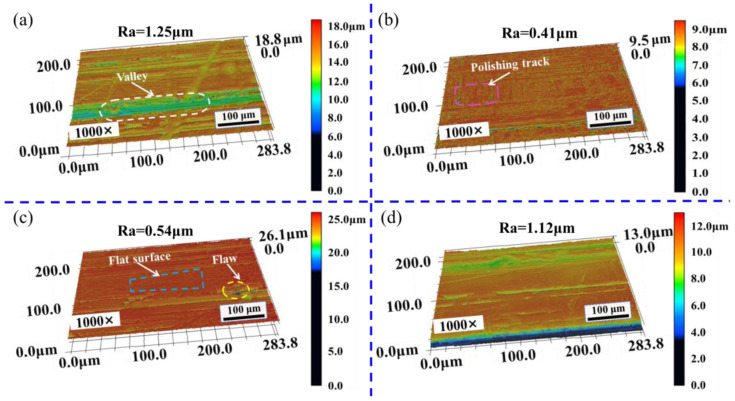
Three-dimensional morphologies of polished surface under different spot overlap ratios after cleaning. (**a**) 99.52%; (**b**) 98.55%; (**c**) 97.58%; (**d**) 96.62% (observed by LSCM at 1000× magnifications).

**Figure 11 materials-16-01477-f011:**
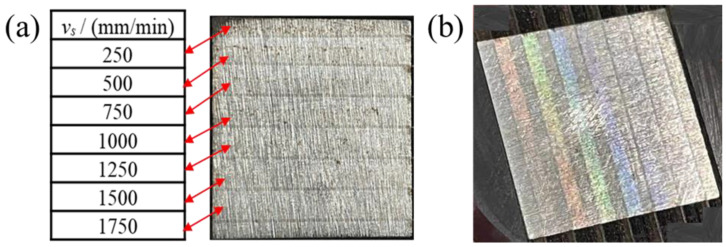
The morphologies of polished surface. (**a**) Macroscopic image showing the coupled laser polished surface; (**b**) Color change of polished surface.

**Figure 12 materials-16-01477-f012:**
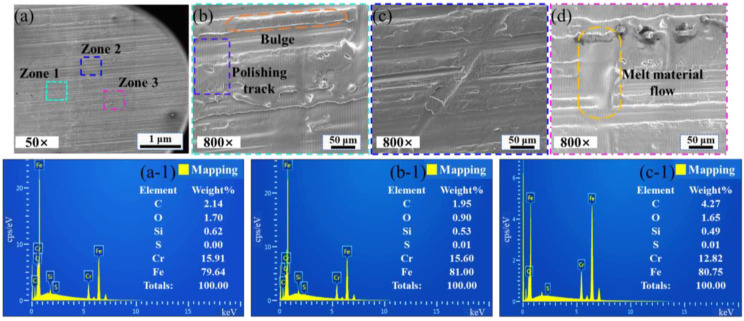
SEM morphologies and element content of polished surfaces with scanning speed of 250 mm/min. (**a**) The polished surface of 50× magnifications; (**b**) Surface morphology of Zone 1; (**c**) Surface morphology of Zone 2; (**d**) Surface morphology of Zone 3; (**a**-**1**) Surface element content on Zone 1; (**b**-**1**) Surface element content on Zone 2; (**c**-**1**) Surface element content on Zone 3.

**Figure 13 materials-16-01477-f013:**
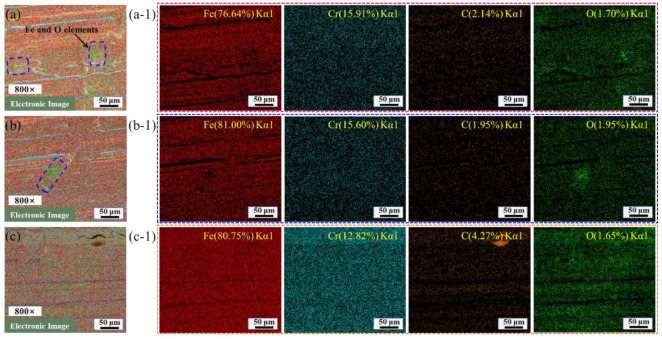
The surface element distribution of polished surfaces with scanning speed of 250 mm/min. (**a**) Zone 1; (**b**) Zone 2; (**c**) Zone 3; (**a**-**1**) Main elements on the surface of Zone 1; (**b**-**1**) Main elements on the surface of Zone 2; (**c**-**1**) Main elements on the surface of Zone 3.

**Figure 14 materials-16-01477-f014:**
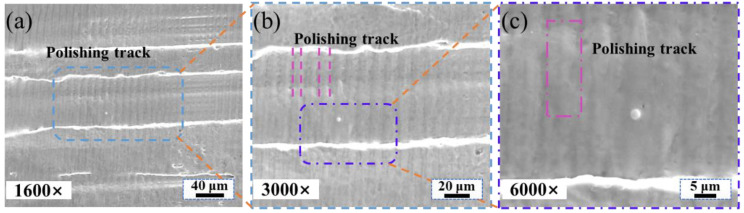
The polished surface with scanning speed of 250 mm/min. (**a**) 1600× magnifications; (**b**) 3000× magnifications; (**c**) 6000× magnifications.

**Figure 15 materials-16-01477-f015:**
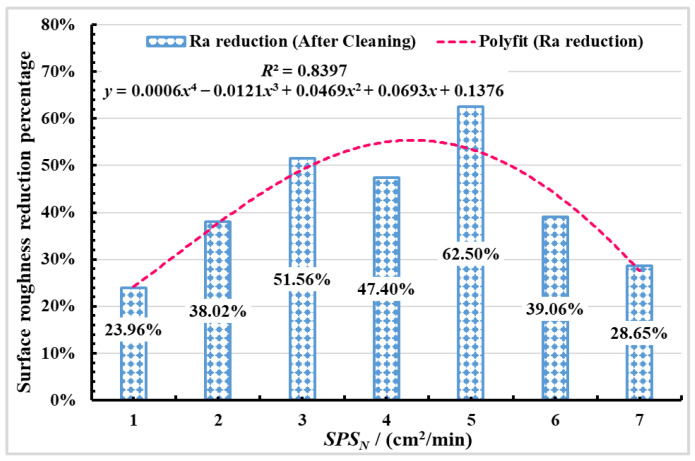
The relationship between the surface roughness Ra reduction percentage and *SPS_N_*.

**Figure 16 materials-16-01477-f016:**
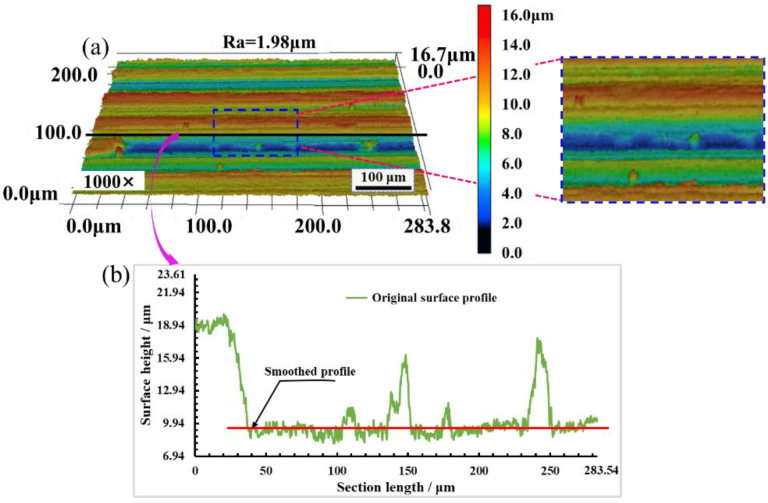
Morphology and profile of original surface. (**a**) Surface morphology; (**b**) Surface profile.

**Figure 17 materials-16-01477-f017:**
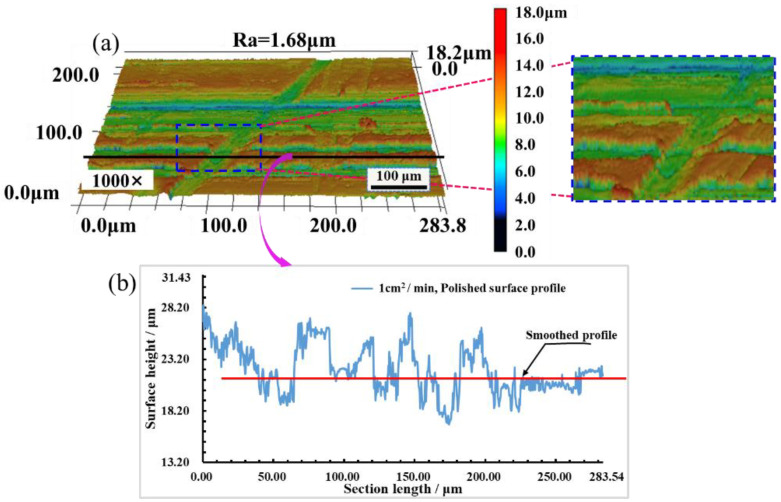
Surface morphology and profile with *SPS_N_* of 1 cm^2^/min. (**a**) Surface morphology; (**b**) Surface profile.

**Figure 18 materials-16-01477-f018:**
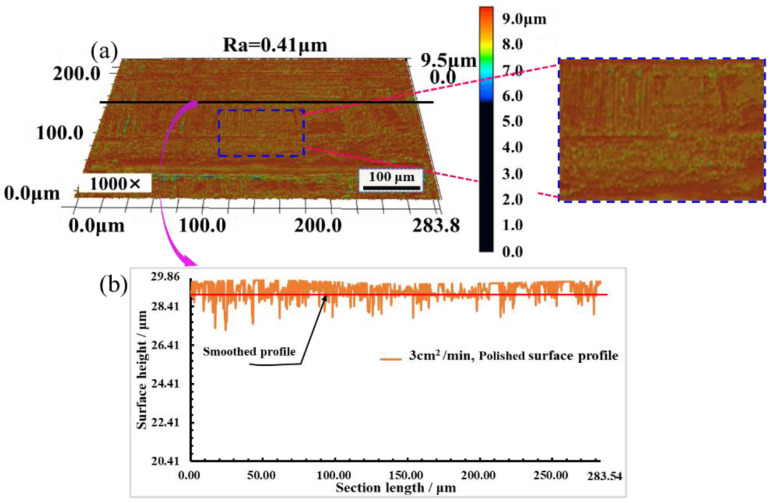
Surface morphology and profile with *SPS_N_* of 3 cm^2^/min. (**a**) Surface morphology; (**b**) Surface profile.

**Table 1 materials-16-01477-t001:** Chemical composition of S136D mold steel (wt.%).

Chemical Element	C	Si	Mn	P	S	Cr	Fe
Mass Fraction (%)	0.32~0.38	0.75~1.05	≤1.0	≤0.025	≤0.001	12.5~14.5	≥83

**Table 2 materials-16-01477-t002:** Detailed parameters of coupled laser polishing operations.

Parameter	Value
*P*/W	1.0
*f*/kHz	10
*w*_0_/μm	43.11
Laser fluence/(J/cm^2^)	1.71
Number of passes	1
Scanning pitch/mm	0.04
Trajectory overlap ratio/%	53.61
*v*_s_/(mm/min)	250~1750

**Table 3 materials-16-01477-t003:** The spot overlap ratios of polished mold steel.

*v*_s_/(mm/min)	250	500	750	1000	1250	1500	1750
Spot overlap ratio/%	99.52	99.03	98.55	98.07	97.58	97.10	96.62

**Table 4 materials-16-01477-t004:** The *SPS_N_* of dual-beam coupled nanosecond laser polishing.

*v_s_*/(mm/min)	250	500	750	1000	1250	1500	1750
*SPS_N_*/(cm^2^/min)	1.00	2.00	3.00	4.00	5.00	6.00	7.00

## Data Availability

Not applicable.

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
