# Peer review of "Effects of Scanning Speed on the Polished Surface Quality of Mold Steel by Dual-Beam Coupling Nanosecond Laser"

_materials, 2023, doi:10.3390/ma16041477_

Round 1

Reviewer 1 Report

The paper proposed a new system for polishing steel surfaces. Even if the research seems to be consistent in the different phases of development, the work needs to be revised before a possible publication. In the following, the main aspects to be revised are reported:

1) page 2, row 93: the authors say "In this paper, a new type of dual-beam 93 coupled nanosecond laser". However, it is not really clear which is the novelty. In par. 2.2, the system is not explained in detail and only the sketch gives an introduction of the setup. Please add more details about the setup, in particular how the second beam is formed and acts. Additionally, according to this indication, please strengthen also the novelty aspect in the introduction section.

2) in the introduction, the authors say "domestic and foreign scholars" (page 2, row 87). What it means? This is a "local" research activity? Please clarify.

3) some repetitions are present, e.g., page 2 , rows 54-55. Please revise carefully all the text before the re-submission.

4) page 5, row 147: what it means "U" shape? it sounds a little bit few scientific....please apply a fit to the data, or talk about presence of a minimum. Otherwise, please add some references about this "U shape"...

5) page 7, row 200: "According to the previous dual-beam ablation experiment". It is not clear which is this previous experiment. Please give an indication. Moreover, the value of the radius is needed also for the above discussion, so please insert this value also in Table 2.

6) at page 9, rows 226-228 the authors tell about the possible presence of oxides. How this presence can influence the applications of the steel surfaces? This could be a problem? Please clarify.

7) par 3.2: the authors introduce the concept of "polishing quality". This is a very important point and needs a more detailed discussione. Please define in the text in the most clear way which are the requirements to obtain a good of polishing quality and if it is possible to quantify a parameter (if any).

Reviewer 2 Report

The manuscript of Huihui Zhang et al. entitled “Effects of scanning speed on the polished surface quality of mold steel by dual-beam coupling nanosecond laser” presents the results of the study of the  scanning speed influence on the efficiency of polishing S136D mold steel using two combined beams of a pulsed laser at a wavelength of 532 nm. The results obtained, most likely, are specific to the test sample with a particular given initial roughness. Therefore, the expediency of publishing this article in its present form is questionable.

               In my opinion, the biggest drawback of the article is that the authors did not demonstrate the advantage of the two-beam method over the single-beam polishing method in their work. It is not clear to the reader what is the mechanism of the two-beam method of material polishing in a particular case, i.e. How do the mechanisms of single-beam and double-beam polishing methods differ from each other?            

In addition, from the optical scheme shown in Fig. 1, it follows that the two-beam method is achieved by splitting the beam of a pulsed laser with a pulse duration of 78 ns. When such laser beams are combined on the surface of the test object, radiation interference should occur (see, for example, Mikheev G. M., Zonov R. G., Kaluzhny D. G. Pulse laser processing of metal thin films on glass substrates / Proc. SPIE., vol. 5399, pp. 179-183, 2004),  which should somehow affect laser polishing. This issue is not discussed in the article.

The article also does not discuss the effect of laser polarization on the polishing process. For example, it is known that the orientation of the polarization of the laser radiation relative to the velocity vector of the laser beam is of great importance in laser cutting of a material.

The article does not provide the value of the angle between beam 1 and beam 2. Does this angle affect the efficiency of laser polishing?

The plane s, on which beam 1 and beam 2 are situated, can be oriented perpendicular or obliquely to the irradiated surface. In addition, the plane s can be parallel and perpendicular to the velocity vector vs (or can make any other arbitrary angle with respect to the vector vs). How does all this affect dual beam laser polishing?

 An important parameter of laser polishing is ?0 beam waist radius (μm). However, there is no value for this parameter in the manuscript.

Authors should improve the presentation of the material. In my opinion, the figure captions are scarce and do not allow the reader to unambiguously understand the information presented.

For example, I do not understand the purpose of the blue rectangular box shown in Fig. 1. In the same figure, for some reason, the beam splitter is called “Spectroscopy Mirror”. Caption to Fig. 2 does not allow one to understand what is drawn in Fig. 2(a). Caption to Fig. 7 does not allow to understand what is drawn in Fig. 7(a): what do ta, tb, tc, td mean? Similar questions arise for other figures.

There are a lot of small comments:

How did the authors measure the laser pulse duration of 78.32 ns with an accuracy of 0.01 ns?

Why are I[A] and U[V] values ​​given in Table 2, which are not defined in the text of the article and which are not used further in the text of the article?

Lines 54 and 54 repeat the same expression "Surface Over Melting (SOM)"; line 85 says "Inconel718" and line 86 says "Inconel 718"; line 87 contains the expression “In summary, domestic and foreign scholars have…..” - for an international journal, the expression “domestic and foreign scholars” is inappropriate; instead of "continuous laser" it is better to use the well-established expression "cw laser"; what does HPDL laser mean (line 83)?; in line 119 "10kHz" should be replaced with "10 kHz"; on the y-axis in Fig. 6 instead of "The spatial energy" should be "The total fluence"; on line 177, "Under" should be "under", and so on.

Based on these comments, I can conclude that the article in this form cannot be published and needs to be substantially improved.

Reviewer 3 Report

In this paper, the authors use a novel dual-beam coupled nanosecond laser to polish mold steel. They studied the effect of scanning speed in a polishing experiment. 

-Please use a white background in all your figures, 

-How can you explain the variability in Figure 4?. 

-Figure 6 and Figure 15 show a trend. Can you please add the determination coefficient and the equation of this trend?

-Please add more discussion about the elements founds on your polished surface.

-Please add a lateral view of your polish in order to explain the heat-affected zones by the speed experimented. 

-Please add more discussion about the traditional molding polishing methods compared with your novel dual beam coupled nanosecond laser.

Round 2

Reviewer 1 Report

After the revision, the paper can be published as it.

Reviewer 2 Report

Response to Reviewer 2 Comments+ reviewer's feedback

Point 1: In my opinion, the biggest drawback of the article is that the authors did not demonstrate the advantage of the two-beam method over the single-beam polishing method in their work. It is not clear to the reader what is the mechanism of the two-beam method of material polishing in a particular case, i.e. How do the mechanisms of single-beam and double-beam polishing methods differ from each other?

Response 1: Compared with single beam, according to the angle between two beams, the spot diameter of the coupling beam is adjustable, the energy distribution at the beam waist is variable, and the spot of coupling beam is elliptical. Different spot diameters of the coupling beam are obtained by adjusting the angle between two beams, which meets the needs of processing capability. With the increase of the angle between the two beams, the eccentricity of the elliptical spot increases, as shown in Page 5, row 173-179.

All this does not justify the advantages of the two-beam polishing method. All of the above-mentioned surface treatment conditions can be obtained with a single beam by controlling the angle of incidence of this laser beam on the surface to be treated. Therefore, the authors should provide the reader with more convincing arguments for the advantages of the two-beam method of surface treatment.

Point 2: In addition, from the optical scheme shown in Fig. 1, it follows that the two-beam method is achieved by splitting the beam of a pulsed laser with a pulse duration of 78 ns. When such laser beams are combined on the surface of the test object, radiation interference should occur (see, for example, Mikheev G. M., Zonov R. G., Kaluzhny D. G. Pulse laser processing of metal thin films on glass substrates / Proc. SPIE., vol. 5399, pp. 179-183, 2004), which should somehow affect laser polishing. This issue is not discussed in the article.

Response 2: The beam emitted by the laser reaches the spectroscope through the 45° reflector mirror, the spectroscope divides the single beam into two laser beams. The beam 1 and beam 2 are coupled into a spot through the focusing mirror. This article mainly studies the influence of variations of the scanning speed of the coupling beam on the surface quality of the mold steel. The mold steel is an opaque material, and the coupling beam can remove the material in the coupling area very well. In the future, it is considered to study the laser radiation interference of coupled beam [30], as shown in Page 3, row 116-118 and 120-124.

If the authors decide to add a new reference [30], it must be properly formatted.

Point 3: The article also does not discuss the effect of laser polarization on the polishing process. For example, it is known that the orientation of the polarization of the laser radiation relative to the velocity vector of the laser beam is of great importance in laser cutting of a material.

Response 3: The beam 1 and beam 2 are coupled into a beam through the focusing mirror, the coupling beam obeys the energy distribution of Gaussian beam, there is no change in the properties of the coupled beam. This paper mainly studies the influence of the scanning speed of the coupling beam on the quality of the polished surface, the influence of laser polarization on the polishing process will be studied in the future, as shown in Page 3, row 127-131.

One can agree with this argument of the authors. However, the authors should present the state of polarizations of the incident laser beams in the manuscript.

Point 4: The article does not provide the value of the angle between beam 1 and beam 2. Does this angle affect the efficiency of laser polishing?

Response 4: The value of the angle between beam 1 and beam 2 is 37.30°. The included angle modulated by the optical path is ideal and can be directly used to study the processing quality of materials. The included angle between the two beams is very small, the included angle affects the laser polishing efficiency, the influence of the included angle on the quality of the polished surface will be considered in the following research, as shown in Page 4, row 131-136.

What does "the included angle" mean? Authors should express their thoughts in a clearer manner.

Point 5: The plane s, on which beam 1 and beam 2 are situated, can be oriented perpendicular or obliquely to the irradiated surface. In addition, the plane s can be parallel and perpendicular to the velocity vector vs (or can make any other arbitrary angle with respect to the vector vs). How does all this affect dual beam laser polishing?

Response 5: This article mainly studies the influence of changing the scanning speed of the coupling beam on the surface quality of the mold steel, and the plane s can be parallel and perpendicular to the velocity vector vs are not the focus of this paper. The effect of velocity vector on coupled beam polishing will be considered in the following research, as shown in Page 5, row 179-182.

One can agree with this argument of the authors. However, the authors should add the orientation of the plane of incidence s relative to the velocity vector vs to the manuscript.

Point 6: An important parameter of laser polishing is ?0 beam waist radius (μm). However, there is no value for this parameter in the manuscript.

Response 6: We have provided the value of waist radius ?0 (μm) in Table 2 on Page 4.

Point 7: Authors should improve the presentation of the material. In my opinion, the figure captions are scarce and do not allow the reader to unambiguously understand the information presented.

Response 7: The presentation of Figure 9 and Figure 10 has been adjusted, Figure 11, Figure 13 and Figure 14 have been added relevant analysis, according to the reviewer’s comments.

Point 8: For example, I do not understand the purpose of the blue rectangular box shown in Fig. 1. In the same figure, for some reason, the beam splitter is called “Spectroscopy Mirror”. Caption to Fig. 2 does not allow one to understand what is drawn in Fig. 2(a). Caption to Fig. 7 does not allow to understand what is drawn in Fig. 7(a): what do ta, tb, tc, td mean? Similar questions arise for other figures.

Response 8: In the blue rectangular box, the beam 1 and beam 2 through the 45° reflector mirror, and then coupled into a spot by a focusing mirror. Following the reviewer's suggestion, we have revised " Spectroscopy Mirror " to " Spectroscope " and indicated it in Fig. 1. Caption to Fig. 2 and Fig. 7 have been changed according to the reviewer’s comments. Schematic diagram of mechanism evolution of SSM, with time sequence of ta, tb, tc, td, according to the reviewer’s comments.

Authors have not answered what do ta, tb, tc, td mean.

Point 9: How did the authors measure the laser pulse duration of 78.32 ns with an accuracy of 0.01 ns?

Response 9: The pulse duration provided by the laser inspection report is 78.32 ns.

I think this is a mistake. I am sure that the pulse duration of the laser used by the authors cannot be specified with such a high accuracy. Therefore, I recommend specifying the pulse duration as 78 ns.

Point 10: Why are I[A] and U[V] values ​​given in Table 2, which are not defined in the text of the article and which are not used further in the text of the article?

Response 10: When the polishing experimental power measured by the dynamometer is 1.0 W, the laser displays the current I[A] and voltage U[V] corresponding to 1.0 W, as shown in Page 4, row 137-138.

The reader is not at all interested in knowing that the laser displays the current I[A] and voltage U[V]. They only want to know the emission parameters of the laser.

Point 11: Lines 54 and 54 repeat the same expression "Surface Over Melting (SOM)"; line 85 says "Inconel718" and line 86 says "Inconel 718"; line 87 contains the expression “In summary, domestic and foreign scholars have…..” - for an international journal, the expression “domestic and foreign scholars” is inappropriate; instead of "continuous laser" it is better to use the well-established expression "cw laser"; what does HPDL laser mean (line 83)?; in line 119 "10kHz" should be replaced with "10 kHz"; on the y-axis in Fig. 6 instead of "The spatial energy" should be "The total fluence"; on line 177, "Under" should be "under", and so on.

Response 11: We have made corrections according to the reviewer’s comments.

Thank you for your valuable comments.
